# Regulatory Processes of the Canonical Wnt/β-Catenin Pathway and Photobiomodulation in Diabetic Wound Repair

**DOI:** 10.3390/ijms23084210

**Published:** 2022-04-11

**Authors:** Sandy Winfield Jere, Nicolette Nadene Houreld

**Affiliations:** Laser Research Centre, Faculty of Health Sciences, University of Johannesburg, P.O. Box 17011, Johannesburg 2028, South Africa; nhoureld@uj.ac.za

**Keywords:** signalling pathways, chronic wound, diabetes, photobiomodulation, Wnt, β-catenin, growth factor

## Abstract

Skin is a biological system composed of different types of cells within a firmly structured extracellular matrix and is exposed to various external and internal insults that can break its configuration. The restoration of skin’s anatomic continuity and function following injury is a multifaceted, dynamic, well-coordinated process that is highly dependent on signalling pathways, including the canonical Wnt/β catenin pathway, all aimed at restoring the skin’s protective barrier. Compromised and inappropriate tissue restoration processes are often the source of wound chronicity. Diabetic patients have a high risk of developing major impediments including wound contamination and limb amputation due to chronic, non-healing wounds. Photobiomodulation (PBM) involves the application of low-powered light at specific wavelengths to influence different biological activities that incite and quicken tissue restoration. PBM has been shown to modulate cellular behaviour through a variety of signal transduction pathways, including the Wnt/β catenin pathway; however, the role of Wnt/β catenin in chronic wound healing in response to PBM has not been fully defined. This review largely focuses on the role of key signalling pathways in human skin wound repair, specifically, the canonical Wnt/β-catenin pathway, and the effects of PBM on chronic wound healing.

## 1. Introduction

Injured or damaged skin is restored via a well-coordinated cutaneous restoration response. The exact molecular and cellular mechanisms behind wound restoration are poorly understood. The most peculiar observation is that repaired skin is different from uninjured skin, largely due to disparities in the processes regulating postnatal cutaneous wound restoration in mammals [1]. Essentially, the restoration response to cutaneous tissue injury includes inflammation, neoangiogenesis, deposition of matrix, and recruitment of cells. A delay in this process is largely coupled with underlying medical conditions including diabetes mellitus (DM), vascular disease, and aging. Non-healing diabetic ulcers are frequently associated with chronicity and limb amputation [2]. Conventional therapies for chronic wounds are available but are limited in their treatment success. Key to the generation of novel therapies for chronic wound management is an understanding of the underlying molecular processes including cellular signalling that is engaged during the wound restoration process [1,2].

A classic wound restoration process (Figure 1) is divided into four succeeding phases, viz., haemostasis, inflammation, resurfacing of new epithelium, and remodelling of connective tissue. These phases are well defined elsewhere [3] and are precisely controlled by intricate communication between cells, signalisation, and extracellular matrix (ECM) proteins [4]. Tissue restoration involves various cell types that go through proliferation, migration, differentiation, and apoptosis aided by various biological signalling pathways [1,5]. The canonical Wnt/β-catenin pathway is implicated in a range of biological activities including cell proliferation, differentiation, and apoptosis and is one of the critical pathways participating in the restoration of cutaneous wounds [6]. Delay in the wound restoration process and the development of wound chronicity are largely due to the reduced production and performance of cytokines and the induction of their specific receptors and intracellular signalling, hindering the functionality of cells including fibroblasts [7].

DM is characterised by elevated blood glucose levels and is a major cause of systemic chronic metabolic disease. Complications related to DM are multiple, involving diabetic neuropathy, retinopathy, and the development of diabetic cutaneous ulceration. Approximately 50–70% of diabetic patients require non-traumatic lower limb amputation due to wound chronicity [8]. The prevalence of diabetic ulcers in Africa is 13%, with about 15% of these necessitating limb amputation, and a mortality rate of 14.2% during hospitalisation [9]. The Wnt/β-catenin pathway is implicated in the appearance of vascular endothelial cells (VECs) in chronic DM, and downregulation of this pathway largely affects the healing of diabetic wounds [10]. 

The treatment of diabetic wounds is frequently challenging, and although there are well-established treatments for diabetic wounds, subsequent reappearance is common in 40% of the treated patients, and approximately 20% of these patients have unhealed wounds 1 year after treatment [11]. Globally, a poor wound restoration response affects many people, as well as healthcare systems, and these challenges call for novel interventions to improve diabetic wound healing. The evolution of technology and improved knowledge regarding the favourable effects of photobiomodulation (PBM) have led to its use as a routine method of treatment for many ailments. PBM employs the non-invasive use of visible and near-infrared (NIR) light (including lasers and light-emitting diodes, LEDs) at definite wavelengths and fluencies [12]. This therapeutic method causes biological changes in cells in response to irradiation. For many years, it has been used in the treatment of psoriasis and jaundice in newborn babies and, more recently, in the treatment of wounds, tissue swelling, pain and inflammation [13]. Studies show that PBM is able to speed up the restoration of chronic wounds and activates cellular proliferation, differentiation and survival by instigating various cellular signalling pathways [12,14].

## 2. Intracellular Signalling in Wound Healing

The cells response to injury is initiated by growth factors and cytokines that play a key role in wound restoration, and their biological action is achieved via signal transduction. Growth factors and cytokines play distinct roles through all phases of wound healing [15]. In response to injury, they can trigger several strategic signalling transduction pathways that are mostly activated during embryonic skin development [1]. Extracellular signal-regulated kinases (ERKs) and calcium (Ca^2+^) are the first intracellular signalling molecules for tissue repair response. These signalling molecules regulate several biological activities including cellular migration, proliferation, contractility, survival and many more related to different transcription factors that are usually induced by several other intracellular signalling pathways. This phenomenon makes it difficult to link a specific signalling response to injury [16]. There is large interaction between intracellular signalling pathways during wound restoration, such as those activated by epidermal growth factor (EGF) [17], transforming growth factor β1 (TGFβ1) [18], Src [19], Ras [20], integrin [21], Wnt/β catenin [5] and Notch [22]. The Wnt/β-catenin intracellular pathway regulates wound healing by improving wound neoangiogenesis, cell proliferation, differentiation, apoptosis and tissue remodelling [6]. 

### 2.1. Wnt/β-Catenin Pathway in Wound Healing

The designation Wnt was created after the name Wingless-linked integration site [23] and identifies a family of glycolipoproteins that regulates embryonic growth and homeostasis in adults. Depending on the type of Wnt ligand, the related signal is via the canonical or non-canonical Wnt signalling pathway. In the canonical Wnt pathway, a co-activator of transcription, β catenin, is the central facilitator (Wnt/β-catenin signalling). Wnt/β-catenin signalling is one of the critical molecular mechanisms for cell proliferation, polarity, determination of fate and tissue restoration. The Wnt/β-catenin signal transduction pathway is blocked when competitive antagonists bind to their specific receptors. Common antagonists of Wnt/β-catenin signalling include Wnt inhibitory factor-1 (WIF 1) and secreted frizzled-related proteins (SFRPs) [24]. Defects in the Wnt/β-catenin pathway are associated with genetic defects, cancer and vascular diseases [25]. 

There are 19 Wnt members in humans, which include Wnt-1, Wnt-2, Wnt-2b, Wnt-3, Wnt 3a, Wnt-4, Wnt-5a, Wnt-5b, Wnt-6, Wnt-7a, Wnt 7b, Wnt-8a, Wnt-8b, Wnt-9a, Wnt-9b, Wnt-10a, Wnt-10b, Wnt-11 and Wnt-16 [26]. Signal transduction in the Wnt/β catenin pathway (Figure 2) begins with the attachment of Wnt proteins to the seven-pass frizzled (Fz) transmembrane receptors and the co-receptor lipoprotein receptor-related proteins (LRP). When the Wnt ligand is not present (OFF), a protein complex consisting of axin, casein kinase (CK) 1, adenomatous polyposis coli (APC) and glycogen synthase kinase 3 beta (GSK3β) is formed. GSK3β causes the phosphorylation of β-catenin, tagging it for degradation by proteasomes. The attachment of Wnt to receptor Fz (ON) advances the stimulation of the dishevelled (Dvl) protein that is responsible for deactivating the axin protein complex. This results in the accumulation of cytoplasmic β-catenin, favouring its translocation to the nucleus and the formation of an active transcriptional complex with T cell-specific factor (TCF) and lymphoid enhancer-binding factor 1 (LEF1) for protein transcription [5,27]. Largely, Wnt3a is involved in activating the canonical Wnt/β-catenin pathway, and in vitro, synthetic Wnt3a activates the Wnt/β-catenin pathway for cell proliferation and differentiation [28].

A large amount of active communication processes occurs in response to injury, eventually leading to wound restoration. An efficacious wound restoration process is largely governed by differentiation and proliferation of various cells including fibroblasts, epidermal stem cells (ESCs) and keratinocytes, achieved through different biological signalling pathways. Incorrect regulation of cellular signalling results in abnormal wound healing, including the development of chronic ulcers. Wnt signalling plays a significant role in controlling cell proliferation, movement and differentiation during tissue restoration [5]. In fibroblasts, the Wnt/β-catenin pathway is inactive and is frequently activated due to injury [29]. Wang et al. (2017) [30] defined a feedback controlling loop joining basic fibroblast growth factor (bFGF) and Wnt signalling via β-catenin in fibroblasts. The bFGF Wnt-regulated pathway is implicated in cell proliferation, and inhibition of bFGF reduces Wnt-mediated influence on cell proliferation. Basic FGF proteins are influential mitogens in normal growth and wound healing [31,32].

### 2.2. Regulation of the Wnt/β-Catenin Pathway in Diabetic Wound Healing

A delay in wound restoration in DM is mainly due to mechanisms related to abnormal inflammation, irregular expression of matrix metalloproteinases (MMPs), reduced cell proliferation, disproportionate cell apoptosis and reduced expression of growth factors and their receptors [3]. High protease levels significantly inhibit dermal reconstruction by reducing ECM components and fibroblast function. Fibroblasts from chronic diabetic wounds are exceedingly senescent, further contributing to reduced ECM deposition [33]. In addition, the reduced healing process in diabetic wounds is worsened by reduced dermal cell neovascularisation, persistent infection and poor cell differentiation within the wound, largely affecting the treatment outcome [28]. The Wnt/β-catenin signalling pathway directly participates in the alteration of various biological processes related to the manifestation and advancement of DM and its complications [24]. 

During diabetic wound restoration, Wnt/β-catenin signalling stimulates skin thickness and pigmentation, and the literature reports that increased regulation of the Wnt/β-catenin pathway augments the action of high-glucose-suppressed cells [6]. It is suggested that reduced activity of the Wnt/β-catenin pathway is due to decreased R-spondin (RSPO) instigated by DM and is one of the main reasons for the irregularity in diabetic wound healing [10]. The RSPO protein family consist of RSPO 1 to 4 secreted proteins that are enhancers of the Wnt signalling pathway. RSPOs are responsible for the stabilisation of the Wnt receptors and their co-receptors via the inactivation of membrane-bound ubiquitin ligases ZNRF3 (zinc and ring finger 3) and RNF43 (ring finger 43) that antagonize the Wnt pathway by targeting the Wnt receptors for ubiquitylation-mediated disintegration [34]. Adjustment or alteration of the Wnt/β-catenin pathway is known to enhance diabetic wound restoration, and it is suggested that transplanting Wnt signalling-activated cells promotes diabetic wound restoration [28,35]. In diabetic wounds, there is a significant decrease in the activity of GSK3β, caspase 3, NF-κB and β-catenin pathways [36]. 

GSK3β, a serine/threonine kinase, is ubiquitously expressed as a strategic regulator of various signalling pathways for cellular proliferation and survival and plays a critical role in phosphorylating the Wnt receptors on LRP5/6, in that way causing stabilization of the Wnt/β-catenin pathway [37]. Inhibition of GSK3β is critical in cell proliferation and differentiation during the wound restorative process, and modulation of GSK3β-mediated Wnt/β-catenin pathway advances diabetic wound healing [38].

## 3. Photobiomodulation (PBM) and the Activation of Signalling Pathways in Diabetic Wound Healing

Many diabetic wounds remain chronic and persistent after a variety of diverse treatments [39]. Various approaches have been employed to hasten the chronic diabetic wound restoration process, and most of these methods involve the use of exogenous stimuli. The use of electrical or electromagnetic energy stimulates the release of growth factors and the enhancement of wound restoration [40]. PBM utilises electromagnetic radiation from low-level monochromatic light to induce non-thermal, photo-chemical and photo-physical effects. The primary and secondary mechanisms of PBM include oxygen- and non-oxygen-related functional pathways activated via energy transfer or multiple non-metabolic pathways for amplified adenosine triphosphate (ATP) production and nuclear gene transcription [41].

PBM is a non-invasive, multipurpose, and economical treatment method that has attracted a lot of attention in the management of chronic wounds [42] and involves illuminating wounds with LEDs or lasers to induce cellular and tissue biochemical activity and wound restoration. In vitro, in distinct animal models and in clinical trials, PBM has been shown to induce wound restoration at unique wavelengths and fluencies, with no ideal set of limitations identified [39]. However, wavelengths between 405 and 1100 nm and a fluence between 0.1 and 10 J/cm^2^ emerge to impart a therapeutic advantage for diabetic wounds, and basically, low-level light has been observed to have a much better effect in accelerating tissue repair than high levels of light [42,43]. De Castro et al. (2020) [44] suggested that PBM enhances the skin wound restoration process, and that the outcomes are directly linked to the selected parameters and the mode of irradiation.

The benefits of PBM in skin wound restoration have been previously reported; however, the biological mechanisms of its action demand to be completely understood. Basically, the target cells photochemically react to illumination via chromophores present in the mitochondria, which absorb the photons. The mitochondrial chromophore, cytochrome c oxidase (COX), unit IV in the mitochondrial respiratory chain, is able to absorb red and NIR light, which results in the increase of different molecules including reactive oxygen species (ROS), ATP, nitric oxide (NO) and calcium ions, and activates several other signalling proteins [45]. PBM is suggested to change the cellular redox environment/state, and several of the significant cellular regulation pathways are redox-mediated. Alterations in cellular redox state activates many intracellular signalling pathways, the synthesis of nucleic acids and proteins, the release of growth factors and the progression of the cell cycle [46]. Oyebode et al. (2021) [47] suggested that cell irradiation produces a biochemical alteration within cells and tissues, inducing cellular processes and diabetic wound restoration. In vitro, irradiated cells show increased expression of many growth factors, including TGF-β and vascular endothelial growth factor (VEGF), illustrative of the PBM-enhanced expression of essential cellular mediators of the wound restorative process [48]. 

Studies have shown that signal transduction pathways facilitate cellular mechanisms of PBM, and many cellular signalling pathways have been shown to be modulated by PBM. However, it is still not completely known which pathways are modulated [49]. Increased intracellular ROS production following PBM influences cellular activities including proliferation, differentiation, migration and survival due to the activation of Src (a non-receptor tyrosine kinase). Src proteins interact with a significant number of intracellular biological signalling transduction pathways including those involving mitogen-activated protein kinase (MAPK), EGF receptor, signal transducer and activator of transcription-3 (STAT3), focal adhesion kinase (FAK) and many others [12]. Several studies testify to the activation and involvement of intracellular signalling during PBM-modulated tissue repair. Feng et al. (2020) [50] suggested that PBM (808 nm; 1 J/cm^2^ and 2 J/cm^2^) enhances wound healing by promoting the migration of gingival stem cells via the ROS/JNK/NF-κB/MMP-1 signalling pathway. Rajendran et al. (2021) [51] noticed decreased oxidative stress through the activation of forkhead Box O1 (FOXO1) in adipose stem cells when PBM (660 nm; 5 J/cm^2^) was used. Rhee et al. (2019) [52] observed reduced neuronal cell polarity and increased cell proliferation, viability and activation of Src, Ras, and MAPK signalling when PBM (660 nm; 0.78, 1.56, 3.12, 6.24, 9.36 J/cm^2^) was used. Ye et al. (2012) [53] used PBM at a wavelength of 632.5 nm and a fluence of 0, 0.6, 1.5 and 2.5 J/cm^2^ and noticed an increase in collagen production and degradation and the activation of Erk1/2 and JNK/MAPK signalling in rat skin. Neves et al. (2018) [54] observed improved acute inflammatory response in the spinal cord of mice, initiated through p38-MAPK when they used PBM at a wavelength of 660 nm and a fluence of 50 J/cm^2^. Bamps et al. (2018) [55] reported increased cellular proliferation via the initiation of AKT, ERK and Ki67 signalling in human head and neck cancer cells when they used PBM at a wavelength of 830 nm and a fluence of 1 J/cm^2^. When Shingyochi et al. (2017) [56] used a CO_2_ laser at a wavelength of 10.6 μm, they suggested that PBM accelerates wound restoration by progressing fibroblast proliferation and relocation via the instigation of AKT, ERK and JNK signalling proteins. In osteoclastogenesis, Song et al. (2021) [57] observed that PBM enhanced the expression of NF-κB, ERK, c-Fos and p38 signalling proteins in osteoclasts. In diabetic fibroblasts, Rajendran et al. (2021) [58] noted mitigated oxidative stress and inhibition of the FOXO1 signalling through the activation of the AKT pathway in diabetic wounded fibroblasts after irradiation at 660 nm with a fluence of 5 J/cm^2^. 

In outer root sheath cells (ORSCs) that maintain the structure of hair follicles, PBM activated proliferation and migration via the Wnt/β-catenin and ERK signalling pathways [59]. Liang et al. (2012) demonstrated the effect of PBM on the AKT/GSK3β/β-catenin pathway and cell apoptosis. They reported a cellular pro-survival effect through the activation of AKT and its interaction and inhibition of GSK3β that led to an increase in cytoplasmic β-catenin and its nuclear translocation [60]. Furthermore, Han et al. (2018) found that a 655 nm LED device promoted hair growth through the activation of Wnt/β-catenin signalling in vitro [61]. The involvement of PBM-initiated Wnt/β-catenin signalling in human fibroblasts in diabetic wound healing is not clear. Presumably, when PBM is used in diabetic chronic wound healing, it utilises the increased mitochondrial ROS, NO and ATP production to initiate the transcription and release of cytokines and growth factors including Wnt proteins. In turn, Wnt binds to Fz receptors, resulting in increased fibroblast activities through the activation of Wnt/β-catenin signalling (Figure 3). In previous studies, it was suggested that PBM (660 nm; 5 J/cm^2^) activated diabetic fibroblast proliferation, migration and survival via the Janus kinase/STAT (JAK/STAT) [62] and phosphatidylinositol 3-kinase/protein kinase B-AKT (PI3K/AKT) [63] pathways. Presently, our research group is studying the involvement of different intracellular pathways including Wnt/β-catenin signalling in the observed diabetic fibroblast activities instigated by PBM.

## 4. Conclusions

Skin injury activates the Wnt signalling pathway and plays an important role in all successive phases of the wound healing process. Studies have suggested several mechanisms, including reduced activity of growth factors and their cell surface receptors, alter the healing process. PBM shows high therapeutic efficacy for different chronic wounds, particularly diabetic skin wounds, and its use has attracted medical interest. The use of PBM as a therapeutic technique for different ailments is rapidly growing. Conversely, the exact means by which emitted light and tissue interact and the parameters that determine the beneficial effects and efficacy of PBM continue to be the main topics of further research. Most importantly, good clinical studies on the molecular, cellular and biologic outcomes of PBM treatment are essential to increase our understanding on the exact means of action of PBM. There is a need for sufficient substantiation to promote the use of PBM.

## Figures and Tables

**Figure 1 ijms-23-04210-f001:**
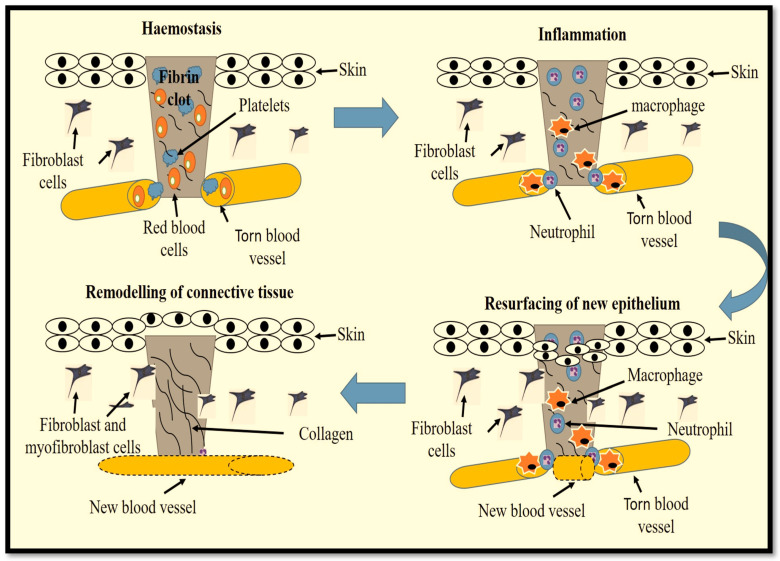
Phases of wound repair and their cellular constituents. The wound restoration process starts with the formation of a platelet plug (haemostasis) to prevent blood loss through the construction of a fibrin matrix. This is followed by removal of debris and prevention of infection (inflammation) by neutrophils and mast cells. Macrophages clear the remaining debris. Fibroblast, epidermal stem cell (ESC) and keratinocyte migration, and neoangiogenesis, aided by various biological signals, contribute to the resurfacing of the epithelium and the formation of granulation tissue and wound closure. The final phase involves remodelling of the connective tissue.

**Figure 2 ijms-23-04210-f002:**
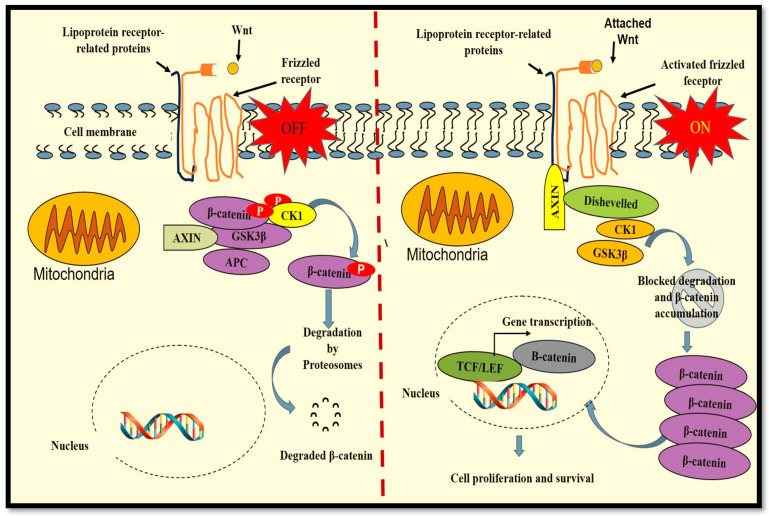
The Wnt/β-catenin pathway. Signal transduction begins with the attachment of Wnt proteins to the seven-pass frizzled (Fz) transmembrane receptors and the lipoprotein receptor-related proteins (LRP). When the Wnt ligand is not present (OFF), a protein complex consisting of axin, casein kinase (CK) 1, adenomatous polyposis coli (APC) and glycogen synthase kinase 3 beta (GSK3β) is formed. GSK3β phosphorylates β-catenin, tagging it for degradation. The attachment of Wnt to Fz (ON) causes the stimulation of the dishevelled (Dvl) protein, resulting in the deactivation of the axin protein complex and the accumulation of cytoplasmic β-catenin. β-catenin translocated into the nucleus forms an active transcriptional complex with T cell-specific factor (TCF)/lymphoid enhancer-binding factor 1 (LEF1) for protein transcription.

**Figure 3 ijms-23-04210-f003:**
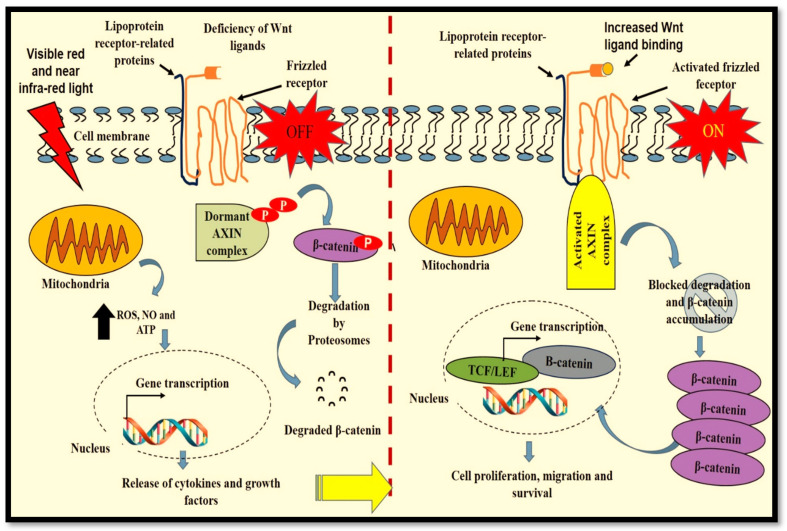
Probable effect of PBM on cellular Wnt/β-catenin signalling in fibroblasts during diabetic chronic wound healing. Wnt/β-catenin signalling is reduced (OFF) in fibroblasts from chronic diabetic wounds. PBM initiates fibroblast increased mitochondrial activity, resulting in increased reactive oxygen species (ROS), nitric oxide (NO) and adenosine triphosphate (ATP), which activates nuclear gene transcription. Increased release of cytokines and growth factors including Wnt ligands activates Wnt/β-catenin signalling (ON) and cytoplasmic accumulation and translocation of β-catenin into the nucleus. In the nucleus, β-catenin forms an active transcriptional complex with T cell-specific factor (TCF) and lymphoid enhancer-binding factor 1 (LEF1) for protein transcription, resulting in cellular proliferation, migration and survival.

## Data Availability

Not applicable.

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
