# Peer review of "Regulatory Processes of the Canonical Wnt/β-Catenin Pathway and Photobiomodulation in Diabetic Wound Repair"

_ijms, 2022, doi:10.3390/ijms23084210_

Round 1

Reviewer 1 Report

The authors did a commendable job in compiling this review. The manuscript is simple, lucid and to the point. The review is timed well much needed now. The authors deftly covered the topics by having good introduction to wound healing and how its abnormalities could be corrected by photo-biomodulation. Wnt/β-catenin pathway in wound healing has been discussed appropriately with clear pictures, they got the reader smoothly to it role diabetic scenario in the following section.  

In the next part they discussed about Photo-biomodulation and how influence different biological pathways of tissue restoration. This part is well written with references of recent research articles. I believe this review is well written and understandable by most of the readers.

Inspite of the best efforts there could be small errors I suggest these minor corrections

Minor comments

Page 1 line 32

“Postponement of this process is largely coupled with underlying medical conditions 32 including diabetes mellitus (DM), vascular disease, and aging”

I suggest the word delay rather than postponement

“Delay in this process is largely coupled with underlying medical conditions 32 including diabetes mellitus (DM), vascular disease, and aging”

The word “postponement” has also been used at page 2 line 47 , Page 5 line 148

I suggest the word “Delay” rather than postponement , Please give a thought about it.

Page 3 line 76

Wavelengths & frequencies not "fluencies"

Reviewer 2 Report

This is a well written review. The articles summaries the basics about the different phases of wound healing, signaling pathways including  Wnt/β-catenin pathway involved in wound healing, followed by dysregulation of the Wnt/β-catenin pathway in diabetic wound healing. Finally, the authors discuss how hotobiomodulation regulates the Wnt/β-catenin pathway in diabetic wound healing.  Here are a few my minor suggestions:

 1: Please cite “de Castro JR, da Silva Pereira F, Chen L, Arana-Chavez VE, Ballester RY, DiPietro LA, Simões A. Improvement of full-thickness rat skin wounds by photobiomodulation therapy (PBMT): A dosimetric study. J Photochem Photobiol B. 2020 Mar 14;206:111850”

2: The abstract needs to add more sentences regarding how hotobiomodulation regulates the Wnt/β-catenin pathway in diabetic wound healing.

3: All Supplementary figures are already included in the paper, why are there the same figures attached as supplementary figures?
